# Herpes zoster in older adults in Ontario, 2002–2016: Investigating incidence and exploring equity

Sarah A. Buchan[1,2,3]*, Nick Daneman[1,2,4,5,6,7], Jun Wang[1,2], Sarah E. Wilson[1,2,3], Gary Garber[1,6,8,9], Anne E. Wormsbecker[10,11], Tony Antoniou[2,10,12,13], Shelley L. Deeks[1,3]

1 Health Protection, Public Health Ontario, Toronto, Ontario, Canada, 2 Populations and Public Health, ICES, Toronto, Ontario, Canada, 3 Dalla Lana School of Public Health, University of Toronto, Toronto, Ontario, Canada, 4 Sunnybrook Research Institute, Toronto, Ontario, Canada, 5 Division of Infectious Diseases, Sunnybrook Health Sciences Centre, Toronto, Ontario, Canada, 6 Department of Medicine, University of Toronto, Toronto, Ontario, Canada, 7 Institute for Health Policy, Management and Evaluation, University of Toronto, Toronto, Ontario, Canada, 8 Ottawa Hospital Research Institute, Ottawa, Ontario, Canada, 9 Department of Medicine, University of Ottawa, Ottawa, Ontario, Canada, 10 Unity Health Toronto, Toronto, Ontario, Canada, 11 Department of Pediatrics, Faculty of Medicine, University of Toronto, Toronto, Ontario, Canada, 12 Li Ka Shing Knowledge Institute, St. Michael's Hospital, Toronto, Ontario, 13 Department of Family and Community Medicine, University of Toronto, Toronto, Ontario, Canada

* sarah.buchan@oahpp.ca

**Data Availability Statement:** The authors do not own nor did they collect the data. I (Sarah Buchan, primary author) have access to the ICES data

## Abstract

Older adults are at increased risk of herpes zoster (HZ) and post-herpetic neuralgia (PHN) and HZ vaccines are available to help prevent infection. The objective of our study was to provide updated data on incidence of HZ and PHN related to clinical and demographic factors in older adults to inform immunization practices. We conducted a population-based, retrospective cohort study and included all cases of HZ seen in outpatient, emergency department, and hospital settings for adults aged 65 years and over between April 1, 2002 to August 31, 2016 in Ontario, Canada. We calculated the incidence of HZ and PHN, and estimated the proportion within each subgroup that developed PHN. We also assessed incidence by neighbourhood-level income quintile before and after the availability of vaccine for private purchase. The average annual incidence of HZ in any setting was 59.0 per 10,000 older adults, with higher incidence in outpatient as opposed to hospital settings. Incidence was higher in the oldest age groups, females, and those classified as immunocompromised or frail. Relative to the pre-vaccine era, the disparities in incidence of HZ by neighbourhood-level income increased, with higher rates of HZ and PHN seen in those residing in lower income quintiles. Additional prevention efforts should be targeted toward adults who are immunocompromised, frail, and those living in lower socioeconomic quintiles. Future work should assess the impact of the zoster vaccine program with a particular focus on equity in the publicly-funded era.

holdings due to having an appointment at ICES which allows me to conduct research at the institution. The dataset from this study is held securely in coded form at ICES. Data sharing agreements and legal restrictions prohibit ICES from making the dataset publicly available in order to protection potentially identifiable personal health information. However, access may be granted to those who meet pre-specified criteria for confidential access, available at www.ices.on.ca/DAS.

**Funding:** This study was supported by Public Health Ontario. Additionally, this study was supported by ICES, which is funded by an annual. Grant from the Ontario Ministry of Health and Long-Term Care (MOHLTC). Parts of this material are based on data and information compiled and provided by: Cancer Care Ontario (CCO), MOHLTC, and the Canadian Institute of Health Information (CIHI). The analyses, conclusions, opinions and statements expressed herein are solely those of the authors and do not reflect those of the funding or data sources; no endorsement is intended or should be inferred. We thank IMS Brogan Inc. for use of their Drug Information Database.

**Competing interests:** The authors have declared that no competing interests exist.

## Introduction

Herpes zoster (HZ) is caused by reactivation of latent varicella zoster virus, related typically to a decline in cell mediated immunity [1]. The incidence of HZ and related complications increases with age, with steep inclines in those ≥50 years [2]. Several studies [3–5], have reported an increase in the incidence of HZ over the last two decades, related possibly to an aging population and/or an increase in the number of individuals living with underlying comorbidities. Immunocompromised individuals are at greater risk of HZ and its complications, including post herpetic neuralgia (PHN) [2, 6, 7], which can have significant impacts on quality of life [8, 9]. Additionally, older frail adults may also be at increased risk [10, 11]; however, this has not been well characterized.

In Canada, two HZ vaccines are currently available. In 2008, a live attenuated (LZV) vaccine was approved initially for those aged ≥65, however, its use is contraindicated in immunocompromised individuals. This vaccine was only available through private purchase from September 2009-September 2016, when it became part of the province of Ontario's publicly funded immunization program for those aged 65–70 years. In October 2017, after the period of the current study, a recombinant subunit vaccine (RZV) was also authorized for use and has been preferentially recommended by Canada's National Advisory Committee on Immunization for those ≥50 years without contraindications but it is currently available only for private purchase in Ontario [12].

The goal of this study was to update and expand our estimates of the burden of HZ and its complications in Ontario, Canada, to inform immunization practices. Specifically, our objectives were to quantify the incidence of medically-attended HZ and PHN in older adults and compare the incidence of HZ and PHN by socioeconomic status (SES) in the privately-funded era of LZV.

## Methods

### Study population, setting, and design

We conducted a retrospective population-based cohort study of Ontario residents aged ≥65 years between April 1, 2002-August 31, 2016. These individuals had universal access to physician services, hospital care, and prescription drug coverage. We selected the start date to align with the introduction of International Classification of Diseases version-10 (ICD-10) codes in Ontario and the end date to coincide with the introduction of the provincial publicly-funded HZ vaccination program for those aged 65–70 years. Our outcome was annual incidence (organized by April to March fiscal year [FY]) of HZ in outpatient, ambulatory, and hospital settings. We obtained ethics approval from Public Health Ontario's Ethics Research Board.

### Data sources and definitions

We used Ontario's administrative health databases, which are securely linked using unique encoded identifiers and analyzed at ICES (www.ices.on.ca). Use of these databases is authorized under section 45 of Ontario's Personal Health Information Protection Act, which does not require patient consent for health system evaluation and improvement. All data were fully anonymized before we accessed them.

**Herpes zoster.** We included an individual's first medically-attended HZ visit during the study period in any outpatient, emergency department (ED), or hospital setting. If an individual was seen in more than one setting in one day, they were classified based on hierarchy of care (i.e. hospital>ED>outpatient). We applied a look-back window to exclude any individuals with HZ in any setting within one year prior to their index date. We used the Ontario

Health Insurance Plan database (i.e. physician billing) to identify individuals seeking outpatient care for HZ. We defined an outpatient visit as anyone with a diagnostic code of 'herpes zoster, shingles' (053) and a prescription for a HZ antiviral (i.e. acyclovir, famciclovir, valacyclovir) dispensed in the five days preceding or following the date of the physician encounter. We excluded individuals with a vaccination fee code (G538, G539) in the same visit to minimize the risk of including those who were vaccinated for HZ as opposed to receiving care for the condition. We used the Ontario Drug Benefit (ODB) database, representing all publicly-funded prescriptions for those ≥65 years who are covered under provincial insurance (all Ontario residents ≥65 are eligible for drug benefits), to identify prescriptions for acyclovir, valacyclovir, or famciclovir. We used the Canadian Institute for Health Information Discharge Abstract Database and the National Ambulatory Care Reporting System to identify hospital and ED visits respectively, combining these outcomes and categorizing them as hospital-attended HZ. We included visits related to HZ (ICD-10 code B02) in any of the 25 allowable diagnostic codes for hospital visits or 10 for ED visits and excluded any case with coexistent varicella (B01). We obtained relevant information on age and sex, as well as population denominators to calculate incidence using the Registered Persons Database, a registry of all Ontario residents with provincial health insurance.

**Covariates.** Using the midpoint of our FY (October 1), we categorized all individuals ≥65 years in Ontario based on a variety of characteristics, including age, frailty status, and immunocompromised status; this allowed us to create annual population-based denominators for each characteristic. Details of our definition for immunocompromising condition have been described elsewhere [13]. Briefly, we included anyone with immune system disorders, HIV, transplant (hematopoietic stem cell transplantation or solid organ transplant), hematologic diseases, recent malignancy and/or ongoing chemotherapy or radiation treatment, and receiving immunosuppressive therapy using a variety of data sources at ICES. We used the Johns Hopkins Adjusted Clinical Groups (ACG) frailty indicator to define frailty [14]. This indicator was designed to identify frailty-defining diagnoses in administrative data and has been used in other studies at ICES [15, 16]. We used neighbourhood-income derived from the Canadian census to classify income quintiles based on residential postal code.

**Outcomes.** We included an individual's first case of medically-attended HZ in hospital, ED, or physician office. We also included acute complications (cutaneous, disseminated, ocular, and neurologic complications associated with HZ) using a window of 7 days before or 30 days after the initial HZ diagnosis in any setting [13]. Only individuals who did not present initially with a complication (i.e. only B02.9 in hospital/ED) and who did not have any acute complication during the follow up time were classified as HZ without complications. We looked separately at the development of PHN in individuals who had HZ, defining the window of PHN diagnosis as 90–365 days after the diagnosis of HZ. We restricted this analysis to individuals who had 365 days of follow-up (i.e. did not die or lose their provincial health insurance eligibility) after their HZ diagnosis date [7]. PHN was defined using a validated algorithm applied to Ontario data, comprising diagnostic codes and prescriptions for drugs used to treat PHN (S1 File) [7, 17]. The maximum follow up for PHN outcomes was March 31, 2017 due to data availability at the time of analysis.

## Statistical analysis

We calculated the annual incidence of HZ per 10,000 older adults (i.e. 65 years and older) in Ontario by age group, sex, income quintile, frailty status, immunocompromised status, as well as age group stratified by frailty status. We calculated these incidences overall as well as by setting, providing separate results for outpatient as well as hospital-attended (hospital or ED)

settings. We also calculated the incidence of HZ with or without acute complications. In our restricted cohort of individuals who had 365 days of follow-up, we calculated the percentage of HZ cases who developed PHN. In a sensitivity analysis, we repeated our analysis in those with 180 days follow-up to determine the proportion that developed PHN. We compared the incidence of HZ and PHN by income quintile in the periods before and after vaccine availability through private purchase (i.e., before and after September 1, 2009). We used negative binomial regression to derive relative risks (RR) and 95% confidence intervals (CI), adjusting for age, sex, and year.

## Results

Between April 1, 2002-August 31, 2016, there were 162,507 incident cases of HZ among older adults; 107,592 (66.2%) of these cases were seen in outpatient settings, 48,193 (29.7%) in the ED, and 6,722 (4.1%) in hospital. A higher proportion of the HZ cases were female (60.7%). Compared to outpatient settings, a higher proportion of adults classified as frail, immunocompromised, or ≥80 years were seen in hospital/ED (Table 1).

The average annual incidence of HZ in any setting was 59.0 per 10,000 older adults, with higher incidence in outpatient as opposed to hospital settings (Table 2). Average annual population denominators during the study period are provided in S1 Table. Incidence was higher in females and increased with age. The incidence of HZ in immunocompromised adults was twice that of those who were immunocompetent (106.5 vs 54.3 per 10,000) and was also higher in those classified as frail, with some variation by frailty within age groups. For instance, the relative impact of frailty was higher in hospital, and in adults aged 65–69 as compared to those aged ≥80, where the incidence did not differ by frailty. Overall, we observed similar incidence in frail adults aged 65–69 as not frail adults aged 70–79. We also observed similar overall incidence by income quintile, but there was a gradient in hospital-attended HZ, with higher incidence in lower income quintiles. These patterns persisted when exploring incidence of HZ with no complications, as well as with any acute complication and PHN (Table 3).

The majority (n = 153,371; 94.4%) of those with HZ were eligible for our PHN analysis (i.e. had 365 days of follow-up). Overall, 21,727 (14.2%) of this restricted cohort developed PHN within 90–365 days after initial HZ diagnosis (Table 4). The pattern of individuals developing PHN was similar to our overall HZ analysis, with a higher proportion of frail, immunocompromised, and adults ≥80 years developing PHN. We observed a higher proportion of frail adults 65–69 who developed PHN relative to those in the same age group who were not frail (21.3% vs 11.2%); however, as age increased, the difference in proportions developing PHN by frailty status decreased. We found similar results in a sensitivity analysis using a 180-day follow-up period.

We explored equity in disease burden by estimating incidence of HZ and PHN by income quintile in two time periods–before and after the availability of LZV in Ontario (Table 5), during which time vaccine was available for private purchase. Prior to licensure, annual incidence of HZ in higher income quintiles was equal to or higher as compared to the lowest income quintile. When the vaccine became available for private purchase, the incidence of HZ was equal to or lower in the higher quintiles as compared to the lowest income quintile, and there was a significant reduction when comparing the highest income quintile to the lowest income quintile (RR = 0.96 [95% CI 0.94–0.98]). When we repeated the analysis with PHN as our outcome, we found a lower risk of PHN in the pre-vaccine era in those in the highest income quintile compared to the lowest quintile (RR = 0.78 [95% CI 0.73–0.83]) with an even greater difference following private availability of vaccine (RR = 0.68 [95% CI 0.64–0.73]).

**Table 1. Characteristics of herpes zoster cases by study setting.**

| | Outpatient, N (%) | Hospital-attended, N (%) | All settings, N (%) |
|---|---|---|---|
| **Overall** | 107,592 (100) | 54,915 (100) | 162,507 (100) |
| **Sex** | | | |
| Female | 66,655 (62.0) | 31,952 (58.2) | 98,607 (60.7) |
| Male | 40,937 (38.0) | 22,963 (41.8) | 63,900 (39.3) |
| **Age group** | | | |
| 65–69 years | 27,169 (25.3) | 13,155 (24.0) | 40,324 (24.8) |
| 70–79 years | 48,356 (44.9) | 23,479 (42.8) | 71,835 (44.2) |
| ≥80 years | 32,067 (29.8) | 18,281 (33.3) | 50,348 (31.0) |
| **Income quintile**[a] | | | |
| Income quintile 1 (lowest) | 20,505 (19.1) | 11,813 (21.5) | 32,318 (19.9) |
| Income quintile 2 | 22,462 (20.9) | 12,035 (21.9) | 34,497 (21.2) |
| Income quintile 3 | 21,091 (19.6) | 10,761 (19.6) | 31,852 (19.6) |
| Income quintile 4 | 21,322 (19.8) | 10,336 (18.8) | 31,658 (19.5) |
| Income quintile 5 (highest) | 21,974 (20.4) | 9,778 (17.8) | 31,752 (19.5) |
| **Frail** | | | |
| Yes | 12,283 (11.4) | 8,015 (14.6) | 20,298 (12.5) |
| No | 95,309 (88.6) | 46,900 (85.4) | 142,209 (87.5) |
| **Age group and frailty** | | | |
| 65–69 years | | | |
| Frail | 1,035 (1.0) | 707 (1.3) | 1,742 (1.1) |
| Not frail | 26,134 (24.3) | 12,448 (22.7) | 38,582 (23.7) |
| 70–79 years | | | |
| Frail | 3,837 (3.6) | 2,527 (4.6) | 6,364 (3.9) |
| Not frail | 44,519 (41.4) | 20,952 (38.2) | 65,471 (40.3) |
| ≥80 years | | | |
| Frail | 7,411 (6.9) | 4,781 (8.7) | 12,192 (7.5) |
| Not frail | 24,656 (22.9) | 13,500 (24.6) | 38,156 (23.5) |
| **Immunocompromised** | | | |
| Yes | 15,183 (14.1) | 11,328 (20.6) | 26,511 (16.3) |
| No | 92,409 (85.9) | 43,587 (79.4) | 135,996 (83.7) |

[a]Numbers may not sum to total due to missing data on income quintile (n = 430 overall)

## Discussion

Over the course of our study period, we observed more than 160,000 incident cases of medically-attended HZ in adults aged ≥65 years. We found higher incidences of HZ in older age groups, females, lower income quintiles, and those who were classified as immunocompromised or frail.

Consistent with published research, the majority of HZ cases were seen as outpatients and a higher proportion of cases were in older adults and females [3, 4, 18, 19]. Our reported incidence was also comparable to other studies, although there are important differences to consider when making comparisons in terms of age groups, setting (inpatient, outpatient), timeframe (including with respect to vaccine availability), and outcome definition. In a systematic review of global incidence, Kawai et al. estimated a rate of 6–8 per 1,000 PY at age 60 and 8–12 per 1,000 PY at age 80 [20]. In several Canadian provinces, incidence of HZ has been reported across a variety of ages and time frames [3–5, 21, 22]. The incidence in our study was lower than that of other studies, including those in Ontario, likely due to our criterion of

**Table 2. Annual incidence of herpes zoster per 10,000 by setting and overall, 2002–2016.**

|  | Outpatient | Hospital-attended | All settings |
|---|---|---|---|
| **Overall** | 39.2 | 19.9 | 59.0 |
| **Sex** |  |  |  |
| Female | 43.7 | 20.8 | 64.5 |
| Male | 33.5 | 18.7 | 52.1 |
| **Age group** |  |  |  |
| 65–69 years | 33.0 | 15.9 | 48.9 |
| 70–79 years | 41.2 | 20.0 | 61.2 |
| ≥80 years | 42.6 | 24.1 | 66.7 |
| **Income quintile** |  |  |  |
| Income quintile 1 (lowest) | 37.8 | 21.6 | 59.5 |
| Income quintile 2 | 39.2 | 20.9 | 60.1 |
| Income quintile 3 | 39.0 | 19.8 | 58.8 |
| Income quintile 4 | 40.1 | 19.4 | 59.4 |
| Income quintile 5 (highest) | 39.9 | 17.6 | 57.4 |
| **Frail** |  |  |  |
| Yes | 40.2 | 25.7 | 66.0 |
| No | 39.0 | 19.1 | 58.2 |
| **Age group and frailty** |  |  |  |
| 65–69 years |  |  |  |
| Frail | 35.8 | 24.1 | 59.9 |
| Not frail | 32.9 | 15.6 | 48.5 |
| 70–79 years |  |  |  |
| Frail | 40.3 | 26.2 | 66.5 |
| Not frail | 41.3 | 19.4 | 60.8 |
| ≥80 years |  |  |  |
| Frail | 41.0 | 25.8 | 66.8 |
| Not frail | 43.1 | 23.6 | 66.7 |
| **Immunocompromised** |  |  |  |
| Yes | 61.2 | 45.4 | 106.5 |
| No | 37.0 | 17.4 | 54.3 |

including an antiviral prescription within five days of an outpatient visit [5, 23]. In British Columbia, Marra et al. reported a decreased incidence in sensitivity analyses restricted to those with an antiviral prescription [3], and we therefore may have underestimated burden.

Across our study period we estimated that 14.2% of older adults diagnosed with HZ developed PHN. A recent systematic review noted the percentage reporting PHN ranged from 5–30%; however, these studies differed in their age groups, study design, years included, and importantly, definition of PHN [20]. Our estimate aligned with several estimates reported in the review among adults >50 years [24–27], although the authors noted that estimates of PHN were often lower when using administrative databases relative to other studies. We adapted an algorithm validated for use in administrative data to identify PHN which has been used in the United Kingdom (UK) [7, 17]. Similar to this study, we found higher risk of PHN in females, immunocompromised adults, and adults living in lower income neighborhoods [7]. Of note, we found that a high proportion of frail adults aged 65–69 developed PHN, which may be impacted by our criteria that those in the PHN analyses had 365 days follow-up. However, the finding remained even after we relaxed the criterion for PHN to 180 days of follow-up.

**Table 3. Average annual incidence of herpes zoster complications[a] per 10,000, 2002–2016.**

| Complications per 10,000 individuals | No complications | Any acute complication (cutaneous, ocular, disseminated, neurologic, other complications) | PHN[b] |
|---|---|---|---|
| **Overall** | 50.4 | 8.7 | 8.2 |
| **Sex** | | | |
| Female | 55.4 | 9.1 | 9.4 |
| Male | 44.1 | 8.1 | 6.7 |
| **Age group** | | | |
| 65–69 years | 42.7 | 6.2 | 5.6 |
| 70–79 years | 52.7 | 8.6 | 8.4 |
| ≥80 years | 55.2 | 11.5 | 10.9 |
| **Income quintile** | | | |
| Income quintile 1 (lowest) | 50.4 | 9.1 | 9.7 |
| Income quintile 2 | 51.2 | 8.9 | 8.7 |
| Income quintile 3 | 50.5 | 8.3 | 8.1 |
| Income quintile 4 | 50.8 | 8.7 | 7.7 |
| Income quintile 5 (highest) | 49.1 | 8.4 | 6.7 |
| **Frail** | | | |
| Yes | 54.1 | 11.9 | 12.1 |
| No | 49.9 | 8.3 | 7.8 |
| **Age group and frailty** | | | |
| 65–69 years | | | |
| Frail | 50.2 | 9.8 | 12.8 |
| Not frail | 42.4 | 6.1 | 5.4 |
| 70–79 years | | | |
| Frail | 55.0 | 11.5 | 12.3 |
| Not frail | 52.5 | 8.3 | 8.1 |
| ≥80 years | | | |
| Frail | 54.4 | 12.4 | 11.9 |
| Not frail | 55.5 | 11.2 | 10.6 |
| **Immunocompromised** | | | |
| Yes | 85.8 | 20.8 | 21.3 |
| No | 46.9 | 7.5 | 7.0 |

*Note.* PHN–post herpetic neuralgia

[a]'No complications' and 'Any acute complication' will sum to total incidence of HZ, but PHN is a separate outcome

[b]Restricted to those who survived 365 days after herpes zoster diagnosis (94.4% of cohort)

## Clinical factors

We estimated the incidence of HZ, acute complications, and PHN in several subgroups, aligning with an increasingly promoted life-course approach to vaccination, in which circumstances beyond age are necessary considerations in making vaccination decisions and in which vaccination to prevent disease is seen as a component in promoting healthy ageing [28, 29]. We found that the incidence of HZ in immunocompromised adults across all settings was twice that of those who were immunocompetent. This is in line with other published estimates but lower than in our previous published work which was limited to hospital-attended HZ [30, 31]; in this publication, we had hypothesized that there may be a greater difference when limited to more severe disease. We have now provided estimates for all medically attended HZ in those ≥65 years by immunocompromised status, which has important implications given that the vaccines available to prevent HZ have different contraindications.

**Table 4. Number and proportion who developed post herpetic neuralgia[a] (between 90–365 days after herpes zoster diagnosis) across study period.**

|  | PHN, N (%) |
|---|---|
| **Overall** | 21,727 (14.2) |
| **Sex** | |
| Female | 13,828 (14.8) |
| Male | 7,899 (13.2) |
| **Age group** | |
| 65–69 years | 4,580 (11.6) |
| 70–79 years | 9,632 (14.0) |
| ≥80 years | 7,515 (16.7) |
| **Income quintile** | |
| Income quintile 1 (lowest) | 5,066 (16.7) |
| Income quintile 2 | 4,821 (14.8) |
| Income quintile 3 | 4,223 (14.0) |
| Income quintile 4 | 3,951 (13.2) |
| Income quintile 5 (highest) | 3,599 (12.0) |
| **Frail** | |
| Yes | 3,171 (18.6) |
| No | 18,556 (13.6) |
| **Age group and frailty** | |
| 65–69 years | |
| Frail | 338 (21.3) |
| Not frail | 4,242 (11.2) |
| 70–79 years | |
| Frail | 1,069 (18.9) |
| Not frail | 8,563 (13.5) |
| ≥80 years | |
| Frail | 1,764 (18.1) |
| Not frail | 5,751 (16.3) |
| **Immunocompromised** | |
| Yes | 4,740 (20.7) |
| No | 16,987 (13.0) |

*Note*. PHN–post herpetic neuralgia

[a]Restricted to those who survived 365 days after HZ diagnosis (94.4% of cohort)

We are not aware of any studies that have provided an estimate of risk of HZ and PHN in adults by frailty status, although it has been suggested that frail older adults may be disproportionately affected [9, 10]. Other research has examined the development of frailty following HZ infection (finding no effect) [32, 33], whereas we assessed the incidence of HZ within cohorts stratified by frailty status and found a higher incidence compared to older adults who are not frail. While aging correlates with frailty, the concept of frailty is distinct and may serve as a better proxy to immunosenescence [34, 35]. This has implications for vaccine promotion given the increase in susceptibility to infectious diseases as the immune response weakens and a goal to target vaccines to those most at risk, beyond considering age alone, but also may have implications for immune response to vaccine [10, 11, 29]. We compared the incidence of HZ and PHN by frailty within age groups in order to better understand the relative contribution of each factor. For both outcomes, we observed a higher incidence in frail older adults within

**Table 5. Incidence of herpes zoster by income quintile, before and after availability of vaccine through private purchase in those ≥65 years[a].**

| | Annual incidence per 10,000 | Unadjusted RR (95% CI) | Adjusted[b] RR (95% CI) |
|---|---|---|---|
| **Herpes zoster** | | | |
| **Prior to private purchase** | | | |
| Overall | 57.6 | | |
| Income quintile 1 (lowest) | 56.8 | ref | ref |
| Income quintile 2 | 58.4 | 1.03 (0.96–1.11) | 1.03 (1.01–1.06) |
| Income quintile 3 | 57.3 | 1.02 (0.95–1.10) | 1.02 (1.0–1.05) |
| Income quintile 4 | 58.6 | 1.05 (0.97–1.13) | 1.05 (1.03–1.08) |
| Income quintile 5 (highest) | 56.9 | 1.02 (0.94–1.10) | 1.02 (1.0–1.05) |
| **After private purchase** | | | |
| Overall | 60.6 | | |
| Income quintile 1 (lowest) | 61.9 | ref | ref |
| Income quintile 2 | 61.9 | 1.01 (0.94–1.07) | 1.0 (0.98–1.03) |
| Income quintile 3 | 60.4 | 0.99 (0.93–1.05) | 0.99 (0.96–1.01) |
| Income quintile 4 | 60.4 | 0.99 (0.93–1.06) | 0.99 (0.97–1.01) |
| Income quintile 5 (highest) | 58.2 | 0.96 (0.90–1.03) | 0.96 (0.94–0.98) |
| **Post herpetic neuralgia** | | | |
| **Prior to private purchase** | | | |
| Overall | 7.8 | | |
| Income quintile 1 (lowest) | 9.0 | ref | ref |
| Income quintile 2 | 8.3 | 0.93 (0.82–1.06) | 0.94 (0.89–1.00) |
| Income quintile 3 | 7.8 | 0.89 (0.78–1.01) | 0.90 (0.85–0.96) |
| Income quintile 4 | 7.3 | 0.84 (0.73–0.96) | 0.85 (0.80–0.90) |
| Income quintile 5 (highest) | 6.7 | 0.76 (0.67–0.87) | 0.78 (0.73–0.83) |
| **After private purchase** | | | |
| Overall | 7.9 | | |
| Income quintile 1 (lowest) | 9.6 | ref | ref |
| Income quintile 2 | 8.4 | 0.88 (0.78–0.99) | 0.88 (0.84–0.94) |
| Income quintile 3 | 7.8 | 0.82 (0.73–0.92) | 0.82 (0.77–0.87) |
| Income quintile 4 | 7.5 | 0.80 (0.71–0.90) | 0.80 (0.75–0.85) |
| Income quintile 5 (highest) | 6.3 | 0.68 (0.60–0.77) | 0.68 (0.64–0.73) |

*Note*. RR–relative risk

[a]Comparison of incidence from September 1, 2009-August 31, 2016 to September 1, 2002 to August 31, 2009

[b]Adjusted for age, sex, year

each age group, but that the relative impact of frailty decreased with increasing age. For example, the incidence of PHN in frail 65–69 year old adults was almost twice that of non-frail adults of the same age group. Conversely, in those ≥80 years, the incidence of PHN was equal by frailty status. As Russell et al. noted, the effect of comorbidity was more pronounced in younger ages [4].

## Socio-economic factors

In addition to clinical factors, we were interested in exploring disease incidence by SES in an effort to understand equity [36]. There is conflicting evidence regarding SES (i.e. income, deprivation) and HZ incidence, with some reporting no association, some reporting an increased risk in more deprived individuals and others reporting an inverse association [2, 37–39]. In

comparison to lower income quintiles, we found similar or greater risk of HZ in higher income quintiles in the pre-vaccine era, but equal or lower risk of HZ in higher income quintiles in the private purchase era. We observed a larger effect when we repeated this analysis with PHN as an outcome. The risk of PHN was significantly lower in the pre-vaccine era in the highest income quintile, but the magnitude of difference between highest and lowest quintiles increased in the private-vaccine era.

There are several reasons for why we may have observed this effect in the two eras. First, only LZV was available in Canada during the study period and its short-term efficacy against PHN has been shown to be higher than efficacy against HZ incidence [25, 38]; as such, the reduced incidence of PHN in particular may suggest that those in the highest income quintile were more likely to purchase vaccine out of pocket. Second, our capture of PHN included a combination of diagnostic codes and prescription data, with prescriptions being limited to those covered by the provincial drug plan. If those in a higher income quintile were being treated with medication not on the formulary, they would not have been identified with PHN and this could lead to a discrepancy in PHN capture between income quintiles. However, other studies have previously reported higher incidence of PHN in low SES prior to the availability of a vaccine [7], and given that we observed a further reduction in PHN in the private vaccine era, these results suggest PHN is lower in those with higher SES. Last, the likelihood of developing PHN may have been impacted by timely access to antivirals or supplemental therapies [40, 41], which could be influenced by SES. Although our PHN analyses were restricted to those with HZ and the majority of individuals were included using criteria that included an antiviral prescription (i.e., outpatients), we did not have data on other medications prescribed or those not publicly-funded.

We are unaware of other studies that have used this approach to identify and understand disparities. In Australia and the UK, HZ vaccine is publicly-funded for older adults. In Canada, Ontario is currently the only province to publicly fund HZ vaccination and does so using LZV for those 65–70 years; however, this program started after our study period [42]. While not comparing disease incidence directly by SES, several studies have reported discrepancies in vaccine uptake by sociodemographic characteristics. In Alberta, Liu et al. estimated vaccine uptake in those ≥60 years for the first four years following LZV licensure and found the odds of vaccination were 57% higher in those in the highest quintile relative to the lowest (OR = 1.57 [95% CI 1.45–1.69]) [43]. In the UK, Ward et al. estimated coverage in 2014–15 as 8.2% lower in the most deprived compared to the least deprived [44]. Also in the UK, Jain et al. reported a lower odds of vaccination in the most deprived compared to the least deprived (OR = 0.69 [95% CI 0.64–0.75]) [45]. Of note, both the UK studies were conducted in an era of public funding, indicating issues related to access may persist beyond cost alone.

## Limitations and strengths

There are several limitations worth noting. First, our study was restricted to those who were ≥65 years. We were concerned that outpatient visits for HZ vaccination could be misclassified in our data as cases of HZ and we therefore added the criteria for an antiviral prescription in conjunction with a coded HZ visit in order to define a case of HZ. However, these prescription data are only available in Ontario for those ≥65 years and we were thus unable to explore incidence in younger age groups. We were also unable to assess those who received prescriptions related to HZ or PHN through private insurance (i.e., prescriptions not be covered by ODB). Further, due to the lack of HZ vaccine-specific billing codes, we could not identify whether an individual had received HZ vaccine. We also did not account for the introduction of varicella vaccine in our analyses, which is a hypothesized contributor to the changing epidemiology of

HZ in adults given the reduction in opportunities for adults to be "boosted" due to lower immunosurveillance [46]; however, results supporting this hypothesis have been conflicting [4, 5, 47–50]. There also may be issues due to misclassification in our data. These include: missing some adults aged 65 who were on prior immunosuppressive therapy because the drug data is only available for those ≥65 years; misclassifying annual immunocompromising status as we defined it based on October 1 each year; and misclassifying ongoing PHN or recurrent zoster as a new case as our washout period was limited to one year.

Our study provides updated burden of disease data for adults ≥65 years in Ontario, and may be a helpful basis for evaluating future immunization programs. We estimated population-based incidence overall and in groups at increased risk of HZ during adulthood in an effort to apply a life-course approach to immunization that incorporates not just age group as an important component of immunization programs but also recognizes that providing protection should take into account other circumstances, such as frailty and immunocompromised [28, 29]. We also provided data on complications, including PHN, which may reduce quality of life and the impacts of which should be taken into consideration when discussing immunization as it relates to healthy ageing.

## Conclusions

Our results suggest that additional prevention efforts should be targeted toward adults who are immunocompromised, frail, and those living in lower income neighbourhoods. Future work should assess the impact of the HZ vaccine program with a particular focus on equity in the publicly-funded era.

## Supporting information

**S1 File. Post Herpetic Neuralgia (PHN) definition.**
(DOCX)

**S1 Table. Average annual population in Ontario aged ≥65 years by characteristic, 2002–2016.**
(DOCX)

## Acknowledgments

We would like to thank Dr. Harriet Forbes for sharing information related to identifying PHN diagnoses in administrative data.

## Author Contributions

**Conceptualization:** Sarah A. Buchan, Nick Daneman, Jun Wang, Sarah E. Wilson, Gary Garber, Anne E. Wormsbecker, Tony Antoniou, Shelley L. Deeks.

**Data curation:** Jun Wang.

**Formal analysis:** Jun Wang.

**Methodology:** Sarah A. Buchan, Nick Daneman, Jun Wang, Sarah E. Wilson, Gary Garber, Anne E. Wormsbecker, Tony Antoniou, Shelley L. Deeks.

**Project administration:** Sarah A. Buchan.

**Writing – original draft:** Sarah A. Buchan.

**Writing – review & editing:** Sarah A. Buchan, Nick Daneman, Sarah E. Wilson, Gary Garber, Anne E. Wormsbecker, Tony Antoniou, Shelley L. Deeks.

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
