## [Decision Letter · Decision Letter 0]

3 Dec 2020

PONE-D-20-28759

Herpes zoster in older adults in Ontario, 2002-2016: investigating incidence and exploring equity

PLOS ONE

Dear Dr. Buchan,

Thank you for submitting your manuscript to PLOS ONE. After careful consideration, we feel that it has merit but does not fully meet PLOS ONE’s publication criteria as it currently stands. Therefore, we invite you to submit a revised version of the manuscript that addresses the points raised during the review process.

We look forward to receiving your revised manuscript.

Kind regards,

Graciela Andrei

Academic Editor

PLOS ONE

Journal Requirements:

2. Given that you are reporting a retrospective study of medical records or archived samples, please ensure that you have discussed whether all data were fully anonymized before you accessed them.

"This study was supported by Public Health Ontario.

 Additionally, this study was supported by ICES, which is funded by an annual. Grant from the

Ontario Ministry of Health and Long-Term Care (MOHLTC). Parts of this material are based on

data and information compiled and provided by: Cancer Care Ontario (CCO), MOHLTC, and the

Canadian Institute of Health Information (CIHI). The analyses, conclusions, opinions and

statements expressed herein are solely those of the authors and do not reflect those of the funding

or data sources; no endorsement is intended or should be inferred. We thank IMS Brogan Inc. for

use of their Drug Information Database."

"This study was supported by Public Health Ontario. The funders had no role in study design, data collection and analysis, decision to publish, or preparation of the manuscript."

Reviewers' comments:

Reviewer's Responses to Questions

**Comments to the Author**

1. Is the manuscript technically sound, and do the data support the conclusions?

Reviewer #1: Yes

Reviewer #2: Yes

2. Has the statistical analysis been performed appropriately and rigorously? 

Reviewer #1: Yes

Reviewer #2: Yes

3. Have the authors made all data underlying the findings in their manuscript fully available?

Reviewer #1: Yes

Reviewer #2: Yes

4. Is the manuscript presented in an intelligible fashion and written in standard English?

Reviewer #1: Yes

Reviewer #2: Yes

5. Review Comments to the Author

Reviewer #1: This retrospective study was meticulously designed. The limitations of it are those seen typically when relying on administrative databases. Bearing this in mind, the authors have made a robust effort to understand the epidemiology of HZ and PHN stratified by demographic, clinical and socio-economic status. The discussion of its limitations and how it might explain differences in incidences compared to previous literature is rational.

The statistical methods used were simple and appropriate for the aim of the study. A mention of the statistical software used would make it complete.

The style of the manuscript is technical and appropriate. If I had one comment to make stylistically, it would be to use subheadings in the 'Discussion' to highlight the differences by 'Demographic', 'Clinical' and 'Social' covariates. This would help to drive home some core messages such as 'Age does not equal frailty.'; 'Differences in SES may relate to cost and/or poor access' and 'Immunocompromise is a significant risk factor with implications for choice of vaccine.'

Reviewer #2: In this paper the authors are reporting the incidence of Herpes Zoster and its complications in Ontario between 2002 and 2016 based on public health records and prescriptions for antiviral drugs commonly used for Herpes Zoster and PHN complications.

Furthermore they want to relate the incidence to equity and access to available vaccines and therefore report the incidence within different percentiles of income in a period where the vaccine only was available at cost price and there was no publicly founded vaccine. This was all based on the assumption that the lower the income the lower the chance that the vaccine was taken.

Unfortunately, they do not show any data to validate this assumption. It is said that due to the lack of HZ vaccine-specific billing codes, they could not identify whether an individual had received HZ vaccine. But in their definition they mention that they excluded people with a coincident vaccination fee code (G538, G539), so apparently there are vaccination fee codes that they could use to calculate the relation between vaccination and income, especially in the VZ era. Of course there could be confounder of the influenza vaccination but that can be corrected for by season or billing for the influenza vaccine. Since there are not very many other vaccines targeting the older population, this could give a good estimate of the assumed relationship between SES and vaccination rate for LZV.

Next to that the data confirm the already known increased risk for Herpes Zoster in frail elderly compared to non-frail elderly, especially in the 65-69 group, immunocompromised patients such as patients using immune suppressive drugs such as patients with autoimmune diseases and transplant patients.

To have a better idea of the impact of targeted Zoster vaccinations it would be nice to have also some estimates about the total size of possible targeted populations, such as immunocompromised patients, or frail elderly. Of course it is possible to calculate population size by combining table 2 incidences with total number of cases from table 1, but it is nice to provide those numbers. With those numbers and the described efficacy of the LZV vaccine to prevent HZ (from literature or from the data suggested above) it would be possible to estimate the cost-effectiveness to target specific candidates with high risk to HZ and its complications. Such additions would certainly improve the impact of the paper and better consolidate the conclusions that additional prevention efforts should be targeted toward adults who are immunocompromised, frail, and those living in lower income neighbourhoods.

6. PLOS authors have the option to publish the peer review history of their article (what does this mean?). If published, this will include your full peer review and any attached files.

Reviewer #1: **Yes: **Dr Alex Chaudhuri

Reviewer #2: No

---

## [Author Response · Author response to Decision Letter 0]

29 Dec 2020

1. This retrospective study was meticulously designed. The limitations of it are those seen typically when relying on administrative databases. Bearing this in mind, the authors have made a robust effort to understand the epidemiology of HZ and PHN stratified by demographic, clinical and socio-economic status. The discussion of its limitations and how it might explain differences in incidences compared to previous literature is rational.

The statistical methods used were simple and appropriate for the aim of the study. A mention of the statistical software used would make it complete.

Thank you for these comments. 

2. The style of the manuscript is technical and appropriate. If I had one comment to make stylistically, it would be to use subheadings in the 'Discussion' to highlight the differences by 'Demographic', 'Clinical' and 'Social' covariates. This would help to drive home some core messages such as 'Age does not equal frailty.'; 'Differences in SES may relate to cost and/or poor access' and 'Immunocompromise is a significant risk factor with implications for choice of vaccine.'

We have included sub-headings in the Discussion section to address this comment (Page 16, line 227; page 17, line 256; page 19 line 296). 

Reviewer #2: 

In this paper the authors are reporting the incidence of Herpes Zoster and its complications in Ontario between 2002 and 2016 based on public health records and prescriptions for antiviral drugs commonly used for Herpes Zoster and PHN complications. Furthermore they want to relate the incidence to equity and access to available vaccines and therefore report the incidence within different percentiles of income in a period where the vaccine only was available at cost price and there was no publicly founded vaccine. This was all based on the assumption that the lower the income the lower the chance that the vaccine was taken. Unfortunately, they do not show any data to validate this assumption.

1. It is said that due to the lack of HZ vaccine-specific billing codes, they could not identify whether an individual had received HZ vaccine. But in their definition they mention that they excluded people with a coincident vaccination fee code (G538, G539), so apparently there are vaccination fee codes that they could use to calculate the relation between vaccination and income, especially in the VZ era. Of course there could be confounder of the influenza vaccination but that can be corrected for by season or billing for the influenza vaccine. Since there are not very many other vaccines targeting the older population, this could give a good estimate of the assumed relationship between SES and vaccination rate for LZV.

In our study, we assessed changes in medically-attended herpes zoster visits using diagnostic codes in outpatient and hospital settings. Our primary outcome of interest was medically attended herpes zoster and one of our analyses included examining incidence before and after availability of vaccine through private purchase. As the reviewer indicates, there are no specific vaccine-billing codes for herpes zoster in Ontario; as such, we could not directly estimate vaccine uptake in our population. However, there are two generic (i.e., no specific vaccine indicated) vaccine fee codes (G538, G539) that are used for a variety of vaccines and in order to prevent outcome misclassification by reducing any chance that someone whose visit was due to a herpes zoster vaccine, and not for herpes zoster itself, we decided to remove any visit with a concurrent vaccination code. We did not use the generic vaccine code as a proxy for herpes zoster as other vaccines that are available to adults over 65 years of age were also captured using the generic code during the study period. This may include pneumococcal vaccine (also funded for adults ≥65 years in Ontario). Further, some practitioners may bill other vaccines such as influenza or tetanus and diphtheria booster [both publicly funded in the province] or travel vaccines using these generic codes. As such, we were unable to use these codes as a proxy for LZV. 

2. Next to that the data confirm the already known increased risk for Herpes Zoster in frail elderly compared to non-frail elderly, especially in the 65-69 group, immunocompromised patients such as patients using immune suppressive drugs such as patients with autoimmune diseases and transplant patients.

Our results provide further evidence about the risk of herpes zoster in immunocompromised patients and older adults; further, the use of frailty as an indicator beyond age alone is a more recent application and one for which quantitative data are not always available. We believe these results will be relevant to policy-makers and practitioners in the context of vaccine decision making. 

3. To have a better idea of the impact of targeted Zoster vaccinations it would be nice to have also some estimates about the total size of possible targeted populations, such as immunocompromised patients, or frail elderly. Of course it is possible to calculate population size by combining table 2 incidences with total number of cases from table 1, but it is nice to provide those numbers. With those numbers and the described efficacy of the LZV vaccine to prevent HZ (from literature or from the data suggested above) it would be possible to estimate the cost-effectiveness to target specific candidates with high risk to HZ and its complications. Such additions would certainly improve the impact of the paper and better consolidate the conclusions that additional prevention efforts should be targeted toward adults who are immunocompromised, frail, and those living in lower income neighbourhoods.

Thank you for this suggestion. We have now included the average annual population size for each characteristic included in Tables 1-4 in our supplementary material (S1 Table).

---

## [Decision Letter · Decision Letter 1]

13 Jan 2021

Herpes zoster in older adults in Ontario, 2002-2016: investigating incidence and exploring equity

PONE-D-20-28759R1

Dear Dr. Buchan,

We’re pleased to inform you that your manuscript has been judged scientifically suitable for publication and will be formally accepted for publication once it meets all outstanding technical requirements.

Kind regards,

Graciela Andrei

Academic Editor

PLOS ONE

Additional Editor Comments (optional):

Reviewers' comments:

Reviewer's Responses to Questions

**Comments to the Author**

1. If the authors have adequately addressed your comments raised in a previous round of review and you feel that this manuscript is now acceptable for publication, you may indicate that here to bypass the “Comments to the Author” section, enter your conflict of interest statement in the “Confidential to Editor” section, and submit your "Accept" recommendation.

Reviewer #1: All comments have been addressed

2. Is the manuscript technically sound, and do the data support the conclusions?

Reviewer #1: Yes

3. Has the statistical analysis been performed appropriately and rigorously? 

Reviewer #1: Yes

4. Have the authors made all data underlying the findings in their manuscript fully available?

Reviewer #1: Yes

5. Is the manuscript presented in an intelligible fashion and written in standard English?

Reviewer #1: Yes

6. Review Comments to the Author

Reviewer #1: Thank you for addressing the reviewers' comments. The paper reads more cogently with all the requisite information clearly presented.

7. PLOS authors have the option to publish the peer review history of their article (what does this mean?). If published, this will include your full peer review and any attached files.

Reviewer #1: **Yes: **Dr Alex Chaudhuri

---

## [Editor Report · Acceptance letter]

2 Feb 2021

PONE-D-20-28759R1 

Herpes zoster in older adults in Ontario, 2002-2016: investigating incidence and exploring equity 

Dear Dr. Buchan:

I'm pleased to inform you that your manuscript has been deemed suitable for publication in PLOS ONE. Congratulations! Your manuscript is now with our production department. 

Kind regards, 

on behalf of

Dr. Graciela Andrei 

Academic Editor

PLOS ONE